# Field Emission Air-Channel Devices as a Voltage Adder

**DOI:** 10.3390/nano10122378

**Published:** 2020-11-29

**Authors:** Wen-Teng Chang, Ming-Chih Cheng, Tsung-Ying Chuang, Ming-Yen Tsai

**Affiliations:** Department of Electrical Engineering, National University of Kaohsiung, Kaohsiung 811, Taiwan; m1085126@mail.nuk.edu.tw (M.-C.C.); a0975852676@gmail.com (T.-Y.C.); frank60755@gmail.com (M.-Y.T.)

**Keywords:** field emission electronic, air-channel, voltage adder, finite element analysis, Fowler–Nordheim tunneling, stress test

## Abstract

Field emission air-channel (FEAC) devices can work under atmospheric pressure with a low operation voltage when the electron channel is far less than the mean free path (MFP) in the air, thereby making them a practical component in circuits. Forward and reverse electron emissions of the current FEAC devices demonstrated symmetric Fowler–Nordheim (F–N) plots owing to the symmetric cathode and anode electrodes. This research aimed to demonstrate the arithmetic application of the FEAC devices, their substrate effect, and reliability. A voltage adder was composed of two FEAC devices whose two inputs were connected to two separate function generators, and one output was wire-connected to an oscilloscope. The devices were on a thin dielectric film and low-resistivity silicon substrate to evaluate the parasitic components and substrate effect, resulting in frequency-dependent impedance. The results show that the FEAC devices possessed arithmetic function, but the output voltage decreased. The FEAC devices were still capable of serving as a voltage adder after the reliability test, but electric current leakage increased. Finite element analysis indicated that the highest electrical fields and electron trajectories occur at the apices where the electrons travel with the shortest route less than the MFP in the air, thereby meeting the FEAC devices’ design. The modeling also showed that a sharp apex would generate a high electric field at the tip-gap-tip, enhancing the tunneling current.

## 1. Introduction

Vacuum field emission (FE) has been applied to numerous fields, such as large-scale accelerators, scanning electron microscopes, FE displays [1], and X-ray sources [2]. Today thermionic vacuum tubes are less used in electrical circuits, but miniature cold FE has applications in vacuum microelectronics and possibly in integrated circuits. Micrometric vacuum microelectronics have found applications in amplifiers [3], logic gates [4], diodes [5], and triodes [6,7]. Vacuum microelectronics provide high transport speed and immunity from radiation owing to the nature of vacuum channels. Vacuum FE devices can also be co-fabricated with a typical Metal-Oxide-Semiconductor Field-Effect Transistor [8]. Minimization of the anode–cathode gap less than the mean free path (MFP) of electrons (200 nm) [9] provides additional benefits to FE devices. FE devices on a nanoscale tunneling transportation can avoid prominent carrier scattering and work under atmospheric pressure [8,9,10,11,12,13,14,15,16]. Second, such devices require a low driving voltage, making them practical in integrated circuits [12,13,16]. Our previous work used an asymmetric field emission air-channel (FEAC) device whose electrodes consisted of multiple triangles and semicircles on the same wafer, which have demonstrated substantially different forward and reverse current characteristics [5]. The results suggest that asymmetric FEAC devices will have potential applications as diodes.

Although FE devices have successfully devised a logic gate and amplifier by connecting two FE devices [3,4], the FE devices required a vacuum environment for measurement because of their microscale transport distance, longer than the mean free path of an electron in the air. This research intended to demonstrate that FE devices are able to perform arithmetic functions in air, i.e., as voltage adders. Fabrication of anode–cathode gaps of FEAC devices should be at a nanoscale distance to avoid scattering. Evaluation of reliability is required to know the feasibility of working as a voltage adder. The research also considered parasitic impedance and substrate effect by fabricating the FEAC devices on a thin dielectric film. The overall impedance should involve the nanoscale anode­–cathode gaps in FEAC devices because the gaps are essentially capacitors [17]. The success of the design should provide a possible application as an electronic component for a harsh environment.

## 2. Experiments

### 2.1. Fabrication of FEAC Devices

The FEAC devices were fabricated at the Taiwan Semiconductor Research Institute. The substrate was an 8-inch p-type (100) silicon wafer. The low resistivity (0.0005–0.001 ohm-cm) substrate resistance can discharge FE current via the SiO_2_ thin film. O_2_ and N_2_O react with the silicon substrate to form a SiO_2_ thin film under dry thermal oxidation at atmospheric pressure in a furnace (ASM A400). The SiO_2_ thin film with a thickness of 15 nm is for dielectric isolation and the discharge capacitor. A tantalum thin film deposition of 100 nm was followed and patterned as a metal FE because metallic electrodes possess efficient field enhancement [17]. This enhancement also depends on the work function of the metals [18,19,20]. Tantalum is a metal with a work function (Φ_M_) of approximately 4.15 eV [21,22] and is commonly used for electrolytic capacitors owing to its high volumetric efficiency (the capacitance in a given volume). Specifically, tantalum as an electrode may generate a relatively high density of electric current. A plasma-enhanced deposition of 30-nm-thick SiO_2_ was utilized for the hard mask before the metal etching process. A layer of negative electron beam resist was subsequently coated. Electron beam lithography formed a pattern whose ends consisted of symmetric tip-to-tip pairs (Figure 1a). Reference [5] has detailed the fabrication process. The measurement suggests that an insulator layer may exist on the electrode interface in the conduction mechanism.

The size of the electrode that consists of the pad and tip area is approximately 0.015 mm^2^. The angle of all the tips is 30°. Although the local electrical field enhances the tips’ sharpness, decrease of the tip angle can also lower the collected current density [23]. This result agrees with our previous studies where the tips with a small angle (20°) exhibited a minimal electric current, while those with a large angle (60°) showed the maximum values [16]. On this basis, this experiment employed a medium angle of 30° tip angle. Next, the electron beam resist was removed, followed by metal etching. The tantalum etching used Cl_2_ and SF_6_ in the transformer coupled plasma reactor (Lam Research 2300). The hard mask was sequentially removed after metal etching. The gaps between the tips measured approximately 40 nm (Figure 1b), whereas the nominal design gap was 50 nm. The doses and exposure time in electron beam lithography were capable of fine-tuning the actual dimensions.

### 2.2. Measurement Setup as a Voltage Adder

A Keithley 2400 source meter was first used to characterize the current–voltage characteristics of the individual FEAC devices in the air. Superimposition of an independent bias voltage on FE devices is a common technique for adjusting FE currents. A bias voltage can optimize a charging current [24], the image resolution or pattern of a scanning electron microscope, and e-beam lithography [25,26,27]. The study applied two independent sources to the two FEAC device inputs and read the addition from an oscilloscope. The experimental setup of a voltage adder used two adjacent FEAC devices (Figure 2a). Two independent functional generators (GW Instek AFG-2105, New Taipei City, Taiwan) generate square waves on the two FE devices via probe manipulators. Figure 2b presents the photo of the FEAC chip mounting on a printed circuit board (PCB). The other electrodes of the two FEAC were joined to form an output and connected to an oscilloscope (GDS-1052-U, New Taipei City, Taiwan). One of the functional generators used an external trigger (ET) to the oscilloscope. Accordingly, the output signal can synchronize with the input signal and observe the output and input waveforms. The oscilloscope itself possesses arithmetic functions, such as adding two waveforms so that the waveforms can be compared with the measured one. The instrument can also give the Fourier transform used in this study. Impedance values of the FEAC devices were measured by an LCR meter (HIOKI IM3533-01, Nagano, Japan) at various frequencies.

### 2.3. Finite Element Modeling

Finite element analysis used COMSOL (5.2, COMSOL Inc., Stockholm, Sweden) to model particle trajectories, potential, and electric field distributions. Particle trajectories and potential distributions used 3D modeling that included an asymmetric field emission device with a thickness of 100 nm on a 15-nm-thick SiO_2_ film. The thickness of the silicon substrate (725 µm) was trimmed to fit a device’s frame. The boundary condition was to apply 5 V on one electrode surface and ground the other (0 V). A hidden box filled with air enclosed the whole device. The separate two layers of 100-nm-thick tantalum on the cathode and anode were for modeling purposes. The lower tantalum film was for performing electron trajectories, and the upper was for applying an electric potential. A 15-nm-thick SiO_2_ thin film was between the electrodes and the silicon substrate. The thickness of a silicon substrate had to be reduced because the silicon substrate was too thick (725 µm) to fit into the modeling frame. The boundary condition was electric potentials of 5 and 0 V (ground) on the two electrodes. A cuboidal box filled with air enclosed the entire device.

## 3. Results and Discussion

### 3.1. Field Emission Properties of the FEAC Devices

The tips’ apices are supposed to exhibit a high electric field so that the significant electron emission should concentrate on the pinnacles. The average tip-to-tip (anode–cathode) gaps (g) of the current FEAC devices at the apices should be smaller than the MFP of electrons in the air to avoid electron scattering. The micrograph in Figure 1b has demonstrated the feature. The measurement of the voltage adder composed of two FEAC devices was under atmospheric pressure.

The current–voltage plot of the current eight-pair tip-to-tip FEAC device exhibits an electrical current in the order of tens of nanoamperes (Figure 3a). The top inset in Figure 3a presents the mechanism of metal–air–metal tunneling. Fowler–Nordheim (F–N) tunneling is a wave-mechanism tunneling of electrons that pass through a triangular energy potential [18]. The barrier height (φ_B_) is between the Fermi level of the cathode and the minimum of the conduction band of the insulator. F–N tunneling occurs when the applied voltage (eV) is higher than the barrier height (φ_B_), while direct tunneling occurs at eV < φ_B_ [28]. The vertical dashed lines in the lower part of Figure 3a demonstrate the range of F–N tunneling in this measurement and thus define the threshold voltages (V_TH_) at the boundaries between F–N and direct tunneling indicated in their F–N plot (Figure 3b). The plots for either side as cathodes are similar because of their symmetric design. The F–N tunneling current (I_F–N_) correlates with the local electric field [18,29], that is, the gradient of potential (V) between the anode and cathode,
(1)$$I_{F\text{–}N}=A\frac{a{\left(\beta V/g\right)}^{2}}{\Phi }\cdot \exp\left(\frac{-b\cdot \Phi^{1.5}}{\beta V/g}\right),$$
where A is the effective area of the cathode, a and b are constants, Φ is the local work function of the emitting surface, and β is the field enhancement factor related to various factors, such as the shape of the cathode and applied voltage [30,31]. The rough surface requires a relatively low voltage to reach the same I_F–N_ level because of its high β [32]. The tunneling current would likely occur at the tip-to-tip instead of two sides because the SEM micrograph presents a relatively smooth sidewall (Figure 1b). Based on Equation (1), the I_F–N_ substantially decreases with the increase of g, that is, I_F–N_ is proportional to 1/g^2^·exp(−C/g), where C is a constant for the same device with a fixed V. The I_F–N_ was even a few orders of magnitude lower with a two-folded increase of g [33]. The ln (I_F–N_/V^2^) presents a negative slope with the reciprocal of the voltage (1/V) for F–N tunneling by taking the logarithm on both sides in Equation (1), that is,
(2)$$\ln(I_{F\text{–}N}/V^{2})=A^{\prime }-\frac{B^{\prime }}{V}$$
where A′ and B′ are the functions related to β. The plot in Figure 3b demonstrates the threshold voltage (V_TH_) is the voltage to convert the tunneling mechanism from direct to F–N tunneling. The V_TH_ transformation between negative and positive slopes of ln (I_F–N_/V^2^) versus 1/V are from 0.5 to 1 V. The low voltage operation makes the current FEAC devices a potential application on electronic components.

### 3.2. Application of the FEAC Devices as a Voltage Adder

Figure 4a presents two input square waves whose frequencies are 100 (red) and 200 kHz (black), the ideal voltage adder (gray) through the addition of two inputs, and the measured output (blue). The output waveform is roughly a square wave, and the amplitude is approximately 1/7–1/6 of the added input. The voltage drop may be attributed to electron scattering under the emission current. The square waves partially distort as the two input frequencies are lowered to 10 and 20 kHz (Figure 4b). A further decrease of the applied frequencies to 1 and 2 kHz exhibits a degraded output voltage (Figure 4c). The discharge time constant is about 186 µS, with a decrease to 1/e, 37% from step response. These results indicate that the FEAC devices possess a filter characteristic that allows high-frequency portions to pass. The maximum output amplitudes also increase with the decrease in frequencies. For example, the maximum output voltage is approximately 250 mV in Figure 4c compared with around 200 mV in Figure 4b. This result may imply additional accumulated charges on the output end when operating at low frequencies compared with high frequencies. Inductive coupling between the two inputs can be an issue because their distance is approximately 200 μm. However, crosstalk contributing to inductive voltage in this voltage adder is less likely an issue because the output voltages deform with decreasing rather than with increasing frequencies. The feature of an inductive voltage is proportional to the operating frequency.

Figure 5 presents the output’s spectral plot in Figure 4a via a fast Fourier transform on the oscilloscope. The cycle time of the output signals in the time domain is 10 μs, i.e., 100 kHz. However, the primary spectrum includes 100 and 200 kHz in the frequency domain. This result indicates that the frequencies of the output signal are traceable from the FEAC devices. Additionally, the noise floor is approximately −100 dBV (10 μV), far lower than the output waveforms.

Our measurement showed that some of the FEAC devices suffered permanent failure due to a vacuum breakdown at excess potential, while the other devices possessed good reliability. A constant electric current (10 nA) stressing was applied to the above-mentioned voltage adder. The test lasted for 10 h in dark conditions. The measured voltages range from 0.26 to 0.28 V, with a sampling rate of 1 s (Figure 6). The electric current stressing applied a current source instead of a voltage source in Figure 3a. The arithmetic function was repeatedly performed after the stressing test. Figure 7a–c presents their ideal output, measured output, and the two input frequencies of 200 and 100 kHz, 20 and 10 kHz, 2 and 1 kHz, respectively, after the stressing test. The figures show that the FEAC devices are still capable of serving as a voltage adder. However, all of the measured voltages (blue) increase substantially. For example, the added voltage in Figure 7a (about 300 mV) increases more than two-fold than in Figure 4a (about 140 mV). This result indicates electric current leakage increases after the reliability test. The growth of the leakage could result from the defect generation in the dielectric layer or the tip-gap-tip.

### 3.3. Frequency-Dependent Equivalent Circuit

Figure 8 presents frequency-dependent impedance from 1 to 100 kHz. Impedance reaches the maximum value of 467 Ω at a critical frequency of 15 kHz. This result implies that the current device exhibits an inductive property below the critical frequency and a capacitive property above the critical frequency. Tip-gap-tip and dielectric isolation capacitance essentially block the DC component of a signal and decline as frequency increases. Parasitic inductance (L_p_) appears based on the previously measured inductive property. The property may result from wire bonds and cables in the low-frequency region. By contrast, the capacitive effect determines the overall impedance under high frequency. The capacitance is from thin SiO_2_ film (C_SiO2_) and tip-gap-tip (C_g_).

Based on the above-mentioned findings, Figure 9 presents an equivalent circuit. The L_p_ dominates impedance under low frequency (<15 kHz) because frequency-dependent impedance of the capacitance (C_SiO2_, C_g_) becomes negligible in the parallel circuit network. By contrast, the capacitance impedance becomes dominant in the circuit network under the high-frequency region (>15 kHz). Although this study considered the substrate resistance (R_sub_) because a low-resistivity substrate is applied, the electric current is mostly from tip-to-tip FE current (i_EF_) rather than underneath substrate current leakage (i_sub_). The current symmetric FEAC devices used the identical substrate to the previous asymmetric FEAC devices that present asymmetric current–voltage curves [5]. The overall impedance (Z_FEAC_) of the parallel network based on Figure 8 is:(3)$$Z_{\mathrm{FEAC}}={j\omega L}_{p}//\frac{1}{{j\omega C}_{g}}//(\frac{1}{{j\omega C}_{\mathrm{SiO}2}/2}{+R}_{\mathrm{sub}})$$
where ω is the angular frequency. The i_EF_ is more significant than i_sub_ [5], so the impedance of the tip-gap-tip capacitor (1/jωC_g_) is significantly smaller than that of SiO_2_ thin film capacitors (1/(jωC_SiO__2_/2) + R_sub_), i.e., the later item is negligible. We may conclude that the maximum impedance occurs roughly at the frequency (f = ω/2π) of
(4)$$f=\frac{1}{2\pi }\sqrt{\frac{1}{L_{p}C_{g}}}$$

Based on Equation (4), the product of L_p_ and C_g_ is about 10^−10^ sec^2^. The tantalum film’s electrical resistance is approximately 1.3 Ω, with an electric resistivity of 131 nΩ-m [34]. This resistance is relatively small compared with that of the 50 Ω BNC cables. C_SiO__2_ is approximately 36 pF by calculating the sandwiched SiO_2_ thin film. The probing electric resistance of the low-resistivity substrate (R_sub_) is about 16 Ω. The discharging time constant (186 μS) is the equivalent capacitance and substrate resistance (C_eq_·R_sub_). This gives the C_eq_ (mainly contributed by C_g_) about 12 µF, which is relatively large compared to the C_SiO__2_ of 36 pF. This result is consistent with the statement that i_EF_ is more significant than i_sub_ and the previous experiment [5].

### 3.4. Finite Element Modeling of the FEAC

Finite element analysis investigated a variety of electrical properties under certain conditions. 3D modeling of the potential distribution shows that the tips’ apices are not equipotential with 5 V on the left and 0 V on the right electrode (Figure 10a). The corresponding particle (electron) trajectory plot shows that the apices as the cathode (right electrode) exhibit a high speed because of the high field (Figure 10b). The results satisfy the design of FEAC devices to operate in the air. Specifically, the electrons primarily emit from the apices where the electron routes are shortest and immune from scattering in the air. The micrograph shows an anode–cathode gap of 40 nm in Figure 1b, far less from the MFP of an electron in the air. The simulation shows that the potential roughly decreases with the distance to the tip, as observed from the color variation. The voltage variation between the anode and cathode converges around the apices. A conventional parallel plate capacitor has a constant capacitance (C) with the charge (Q) over the applied potential (V), that is, the electric field is constant except for the fringe. Though the current FEAC device is commonly regarded as a capacitor, the simulation shows a substantial potential variation (ΔV) at the tip-gap-tip. The electric charge is dynamic due to the nature of FE devices. The time-independent bias makes the conductance (*g_m_*) of the device correlate to the change of the capacitance, that is,
(5)$$g_{m}=\frac{i(V)}{V}=C\frac{\mathrm{dV}}{\mathrm{dt}}+V\frac{\mathrm{dC}}{\mathrm{dt}}$$

The exponential increase of electric current with voltage (Figure 3a) implies that the FEAC device’s capacitance (C_g_) significantly increases with the applied voltage. By contrast, a parallel plate capacitor has a constant capacitance without considering the FE and fringing effect.

The geometric tips serving as electron emission would also influence the local electric field distribution. Figure 11a,b demonstrates electric field distribution along the *x*-axis (E_X_) with respective sharp and blunt tips, enlarged from the modeled FEAC devices in the insets of Figure 10a,b. It only compares two devices’ E_X_ because the thicknesses (*y*-axis) and angles (related to the *z*-axis) of the two figures’ triangular tips are the same. The simulation shows that the sharp tip exhibits a significant E_X_ difference at the tip-gap-tip than the blunt one. The result implies that pointed apices are a required design to enhance the electric field at the tip-gap-tip, which agrees that the rough surface usually possesses a high enhancement factor [32,33].

## 4. Conclusions

This study characterizes the multiple tip-to-tip FEAC and investigates a voltage adder’s application with two FEAC devices. The experimental design enables the FEAC devices to operate in the air with a minimal anode–cathode gap at the apices less than the electrons’ MFP to make them immune to electron scattering. The fabrication results agreed with the design; thus, the devices were able to work at atmospheric pressure and to serve as a voltage adder. The results show that the FEAC devices can work in the F–N tunneling region at less than 1 V with an approximately 40-nm anode–cathode gap. The finite element modeling indicates the high electric field and acceleration of electrons around the apices, which agrees with the current design’s purpose. In addition to the tip-to-tip distance, the roughness and the angle of the tips above-mentioned can determine their tunneling current. The finite element modeling also shows the tips’ sharpness to change the electric field distribution at the tip-gap-tip. The performance and the reliability test prove that the FEAC device has potential applications as an electronic component.

## Figures and Tables

**Figure 1 nanomaterials-10-02378-f001:**
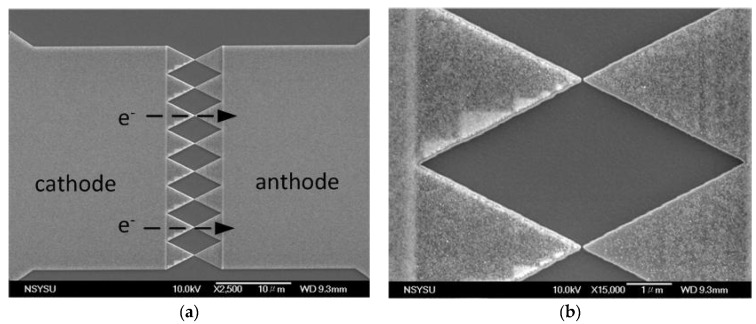
(**a**) Micrograph of an eight-pair tip-to-tip pattern as a field emission air-channel (FEAC) device, allowing electrons (e^−^) to flow from cathode to anode; (**b**) the tip-to-tip gaps are approximately 40 nm.

**Figure 2 nanomaterials-10-02378-f002:**
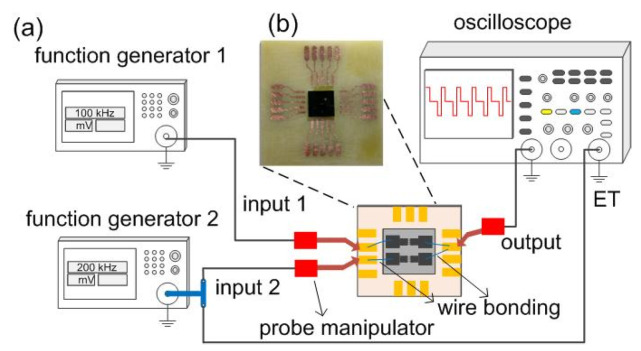
(**a**) Experimental setup using two function generators as the two inputs on the two FEAC devices. The other two electrodes of the FEAC devices are merged as the output that connects to an oscilloscope. A chip is mounted on a printed circuit board, and the probe manipulators connect the FEAC devices and the instruments. One of the inputs is connected to the oscillator’s external trigger (ET) to synchronize the output with the input signal. (**b**) Photo of the FEAC devices mounted on a printed circuit board.

**Figure 3 nanomaterials-10-02378-f003:**
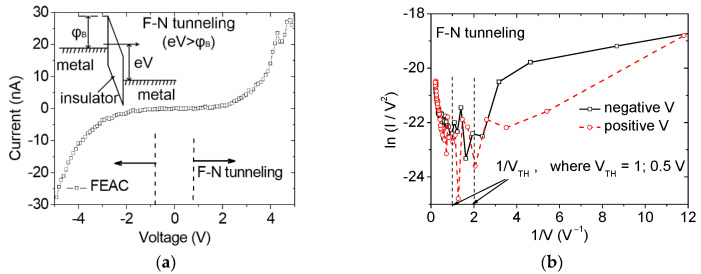
(**a**) Linear current–voltage plot and the schematic of energy band for F–N tunneling of metal–air–metal (top inset), and (**b**) F–N plots for the positive and negative voltage parts indicating that the threshold voltage to become F–N tunneling is between 0.5 and 1 V.

**Figure 4 nanomaterials-10-02378-f004:**
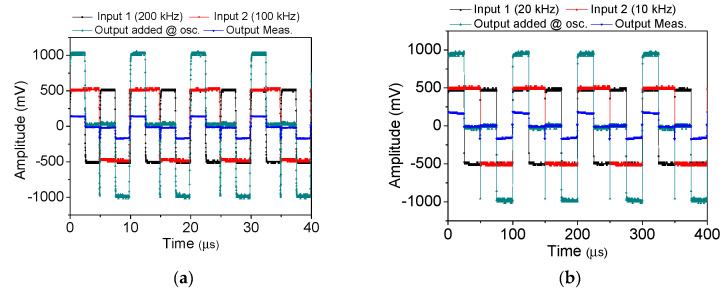
Output measurement (blue), added inputs (gray), and two square wave inputs (black and red) with distinct frequencies of (**a**) 200 and 100 kHz, (**b**) 200 and 100 kHz, and (**c**) 2 and 1 kHz.

**Figure 5 nanomaterials-10-02378-f005:**
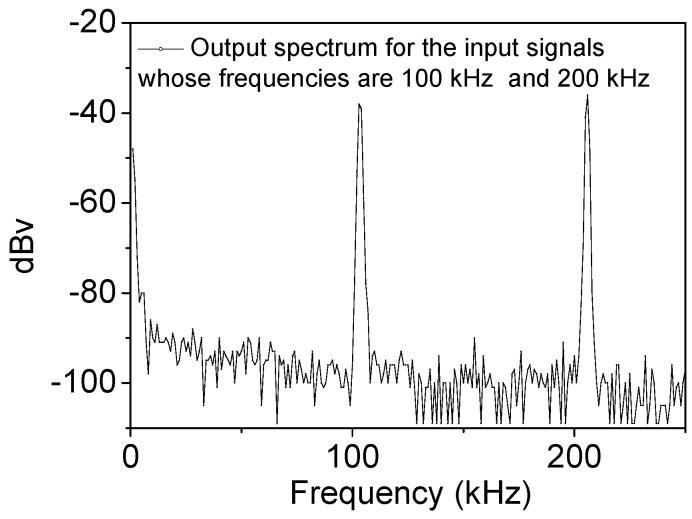
The spectral plot of the output with two input frequencies of 100 and 200 kHz.

**Figure 6 nanomaterials-10-02378-f006:**
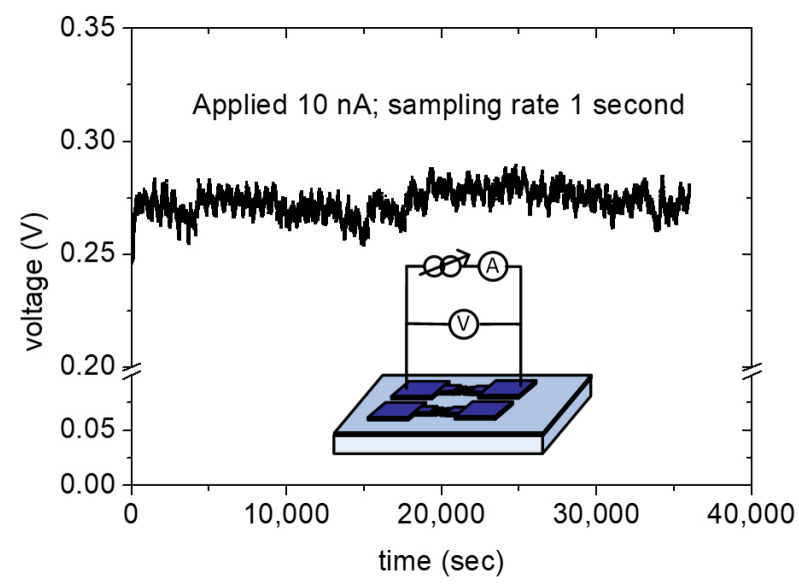
The measured voltages with the electric current stressing of 10 nA to one of the FEAC devices for 36,000 s (10 h).

**Figure 7 nanomaterials-10-02378-f007:**
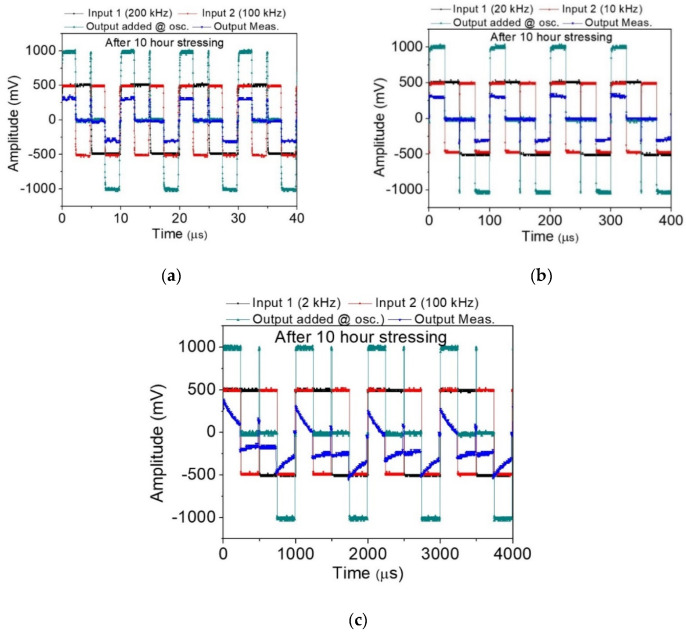
The voltage adder of reliability of post-test after 10-h electric current stressing. The output measurement (blue), added inputs (gray), and two square wave inputs (black and red) with distinct frequencies of (**a**) 200 and 100 kHz, (**b**) 200 and 100 kHz, and (**c**) 2 and 1 kHz compared to pre-test in Figure 4.

**Figure 8 nanomaterials-10-02378-f008:**
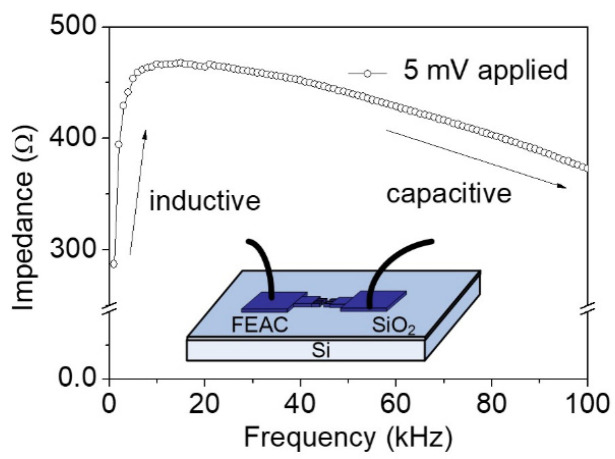
Impedance measurement value with frequency from 1 to 100 kHz showing the maximum value at approximately 15 kHz, thereby implying inductive reactance before the frequency and capacitive reactance after the frequency.

**Figure 9 nanomaterials-10-02378-f009:**
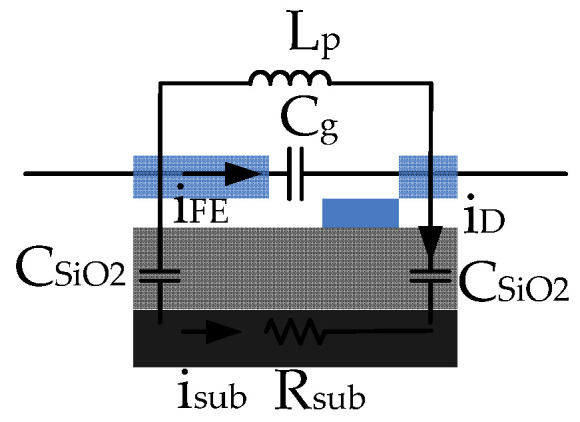
Schematic of the proposed equivalent circuits considering the modeling elements of the tip-gap-tip capacitor (C_g_), thin SiO_2_ film capacitors (C_SiO2_), substrate resistor (R_sub_), and parasitic inductors (L_p_). The discharge current (i_D_) happens after the tunneling current flows through the tip-gap-tip gap (i_FE_) and substrate current leakage (i_sub_).

**Figure 10 nanomaterials-10-02378-f010:**
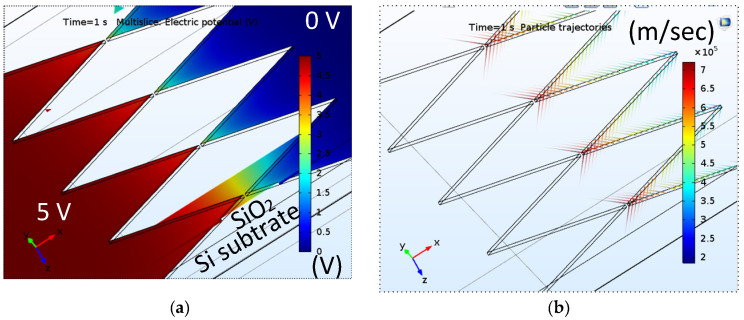
(**a**) Electric potential distribution on a tip-to-tip FEAC device by 3D finite element modeling, where the left electrode is 5 V, and the right is 0 V. The cross-section indicates the electric potential distribution along the *x*-axis. The color range spans from 0 to 5 V; (**b**) particle trajectories indicating the speed of electrons by colors due to acceleration by an electric field.

**Figure 11 nanomaterials-10-02378-f011:**
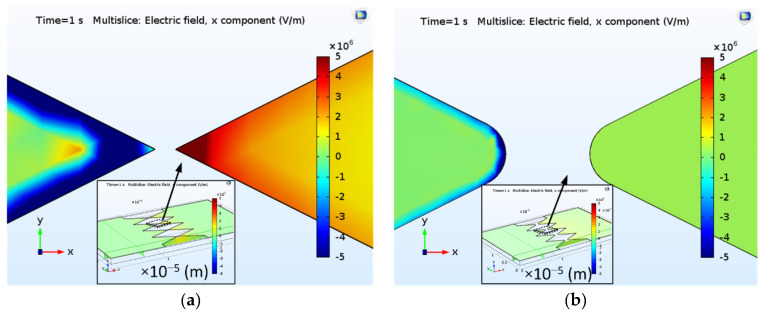
Electric field distribution along the *x*-axis (E_X_) with (**a**) a sharp and (**b**) a blunt tip, enlarged from their tip-gap-tip portions of the 3D models (insets). Note that all of the color bar ranges are the same (E_X_: from −5 × 10^6^ to 5 × 10^6^ V/m). The gradient of the electric potential (E_X_) is along the *x*-axis. The |E_X_| increases gradually toward the tips. The negative or positive E_X_ depends on the low and high potential on the two electrodes.

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
