# Peer review of "Field Emission Air-Channel Devices as a Voltage Adder"

_nanomaterials, 2020, doi:10.3390/nano10122378_

Round 1

Reviewer 1 Report

This manuscript reports on an investigation of a field emission devices operating at atmospheric pressure and demonstrates voltage addition between two such devices. After describing the fabrication of the devices and the setup used for the voltage addition measurements, the authors characterize the current-voltage response of the device and show that it is able to operate in the Fowler-Nordheim tunneling regime for applied voltages above 0.8 V. Voltage addition of low frequency square signals and the device impedance are then measured in the 1-100 kHz range. The authors then perform FEM simulations of the field emission effects in the structure.

The topic is in principle one of interest for a journal like Nanomaterials and the realization of FEAC devices operating in air is worth investigating. However, I find the reported work to be on the "light" and superficial side, both in terms of presentation and results. In particular, the "modeling" (effective circuit, FEM simualtions) does not bring anything interesting to complement the experimental observations. In addition, the quality of english and the clarity of the text is very low. In my opinion much work is required before this study reaches a level that is acceptable for publication, in Nanomaterials or elsewhere.

Below is a list of issues--big and small--that should be addressed
-the abstract is very unclear. What is general and what is specific to the present work? What is the system considered? What are (quantitatively) the performances of the devices? etc.
-the introduction does not properly compare the devices of interest with other state-of-the-art devices, making it difficult to judge the importance and relevance of the presented results.
-it is not clear how the threshold voltage of 0.8V is calculated from the data of Fig. 3b.
-the discussion of the frequency dependecy on page 5 (between Figs. 3 and 4) is confusing. For instance, what do the authors mean with "the crosstalk can be negligible because the voltage change with frequency is insignificant"? Do they mean phase or amplitude of the measured output voltage? Amplitude response, charging effects could be discussed further.
-the purpose of the FT spectrum of Fig. 5 is unclear. What do the authors conclude from this measurement?
-Section 3.3 introduces an equivalent circuit, but no attempt at extracting quantitative information on the components is made from the measurements. The authors only state the obvious that there are inductive and capacitive components!
-The same remark can be made from the FEM simulations of Sec. 3.4. No useful quantitative information on effective/parasitic capacitance, inductance, impedance, etc. is extracted. In addition, how the simulations are performed is not clear. Neither is the presentation of the results (why is there a "rainbow" in the vicinity of the tip pair at the edge on Fig. 8a? what do the authors mean with "Delta V" and "Delta Q"?

Reviewer 2 Report

This manuscript shows a FEAC device using a tip-to-tip metal pattern and an adder as an application. The overall experiment and logic are good, but the presented device only shows normal results. In particular, it is questionable whether the content of the manuscript matches the Aims & Scope of this journal. For these reasons, I do not suggest the publication of nanomaterials. (I recommend to transfer to another journal)

I offer additional suggestions to improve this manuscript.

  • In order to intuitively check the structure of the device, it is recommended to provide a brief scheme or insert text into the SEM image in Figure 1(a).
  • Please add a description of the role and function of Output add and Output meas. in Figure 4. And Match the color of each data in Figure 4. (In Fig. 4(a), input 1 is black, but in Fig. 4(c), it is red.)
  • In Figure 4(c), there is no result for output meas. Why isn't this data missing at low frequencies?
  • (line 182, page 6) The inductive characteristic occurs in the structure of the coil. How do inductive components occur in thin metal films?
  • (line 207, page 7) please check the sentence.

Reviewer 3 Report

The author reported the field emission air-channel (FEAC) devices working under atmospheric pressure with a low operation voltage. Moreover, a voltage adder was realized by using 2 FEAC devices. Finite element analysis was also employed to study the electrical field and electron trajectories in the FEAC device.

Please discuss more about the variation of performance of the FEAC devices, if there is any, for the small variations of in the geometry of the device during the fabrication. For example, the small variation in the distance of tip-to-tip gap, the surface roughness of the tantalum tip. Providing more information about the repeatability/standard deviation of the exact geometry and roughness (Figure 1) will help the reader to understand more about this device.

For figure 4, will be good if the author can provide the rough lifetime of this device, that is to mention if the characteristics of this charging and discharging cycles will be degraded after a long time, since the result only show the characteristic within 1 s. This data helps to understand the feasibility for the FEAC to work as a voltage adder, repeatedly.

Please give explanation to the difference in the electric field distributions observed in Figure 9 for the different gap. In the lowest tip-to-tip gap, the electric field is strongest at the two apices. However, for the about 3 tip-to-tip gaps, the electric field is much strong in between the gap then at the two apices.

Round 2

Reviewer 1 Report

The authors have appreciably revised their mansucript. The presentation is now clearer and the analysis mroe convincing. It would in my opinion now be suitable for publication in Nanomaterials.

Author Response

The manuscript has been proof-read and eliminated grammatical errors.

Reviewer 2 Report

I have checked the revised manuscript.
Most parts have been made clearer.
Therefore, I think that this manuscript can be published in Nanomaterials.

Author Response

(The authors gave the same response as above.)

Reviewer 3 Report

Thanks the author for looking into my comments.

Author Response

(The authors gave the same response as above.)
